# Autophagy Inhibition Induces the Secretion of Macrophage Migration Inhibitory Factor (MIF) with Autocrine and Paracrine Effects on the Promotion of Malignancy in Breast Cancer

**DOI:** 10.3390/biology9010020

**Published:** 2020-01-18

**Authors:** Israel Cotzomi-Ortega, Arely Rosas-Cruz, Dalia Ramírez-Ramírez, Julio Reyes-Leyva, Miriam Rodriguez-Sosa, Patricia Aguilar-Alonso, Paola Maycotte

**Affiliations:** 1Centro de Investigación Biomédica de Oriente, Instituto Mexicano del Seguro Social, Km 4.5 Carretera Atlixco-Metepec HGZ5, Puebla 74360, Mexico; israelcotzomi@gmail.com (I.C.-O.); arelyrosascruz@gmail.com (A.R.-C.); dal2_r@hotmail.com (D.R.-R.); julio.reyes@imss.gob.mx (J.R.-L.); 2Facultad de Ciencias Químicas, Benemérita Universidad Autónoma de Puebla, Ciudad Universitaria, Puebla 72570, Mexico; patricia.aguilar@correo.buap.mx; 3Unidad de Biomedicina (UBIMED), Facultad de Estudios Superiores Iztacala (FES-I), Universidad Nacional Autónoma de México (UNAM), Tlanepantla 54090, Mexico; rodriguezm@unam.mx; 4Consejo Nacional de Ciencia y Tecnología (CONACYT)—Centro de Investigación Biomédica de Oriente (CIBIOR), Instituto Mexicano del Seguro Social (IMSS), Puebla 74360, Mexico

**Keywords:** autophagy, MIF, breast cancer, reactive oxygen species, ROS, chloroquine, secretion, p115

## Abstract

Breast cancer is the main cause of cancer-related death in women in the world. Because autophagy is a known survival pathway for cancer cells, its inhibition is currently being explored in clinical trials for treating several types of malignancies. In breast cancer, autophagy has been shown to be necessary for the survival of cancer cells from the triple negative subtype (TNBC), which has the worst prognosis among breast cancers and currently has limited therapeutic options. Autophagy has also been involved in the regulation of protein secretion and, of importance for this work, the inhibition of autophagy is known to promote the secretion of proinflammatory cytokines from distinct cell types. We found that the inhibition of autophagy in TNBC cell lines induced the secretion of the macrophage migration inhibitory factor (MIF), a pro-tumorigenic cytokine involved in breast cancer invasion and immunomodulation. MIF secretion was dependent on an increase in reactive oxygen species (ROS) induced by the inhibition of autophagy. Importantly, MIF secreted from autophagy-deficient cells increased the migration of cells not treated with autophagy inhibitors, indicating that autophagy inhibition in cancer cells promoted malignancy in neighboring cells through the release of secreted factors, and that a combinatorial approach should be evaluated for cancer therapy.

## 1. Introduction

For women, breast cancer is the most commonly diagnosed type of cancer and the leading cause of cancer-related deaths in the world [1]. Breast cancer is a highly heterogeneous disease, molecularly defined by gene expression signatures and clinically classified by immunohistochemical tests for hormonal (estrogen and progesterone) and human epidermal growth factor receptor 2 (HER2), receptors which have prognostic value and are used for defining targeted therapies [2]. Among breast cancer subtypes, triple negative breast cancer (TNBC) is characterized by the absence of hormonal (estrogen and progesterone) and HER2 receptors, and thus has no targeted therapy. TNBC accounts for 15–20% of breast cancers and has an aggressive phenotype, with earlier onset of metastatic disease, visceral metastases, rapidly progressive disease, short response duration to available therapies, and shows the worst prognosis among all breast tumors in every stage of the disease [3,4]. Numerous clinical studies are investigating novel therapies for TNBC, including immune-checkpoint inhibitors, PARP-targeted therapies or androgen-receptor antagonists [4]. Also, due to the high degree of genomic instability and aneuploidy present in TNBC [5], as well as specific signaling pathways activated in this subtype of cancer, other possible targets have been suggested, including cellular quality control mechanisms like the proteasome [6] and autophagy [7].

Macroautophagy, hereafter autophagy, is a cellular homeostatic process in which cytoplasmic material (proteins, organelles) is sequestered in a double-membrane structure named the autophagosome, which later fuses with the lysosome for the degradation of its content [8]. At basal levels, autophagy serves a homeostatic function operating as an intracellular quality control system, and alterations in the autophagic pathway have been related to pathological processes like neurodegenerative, metabolic, infectious diseases and cancer [8,9]. Autophagy can also be induced at higher levels in response to diverse stimuli like nutritional, metabolic, oxidative, pathogenic, genotoxic and proteotoxic stress, often serving a cellular survival function [8]. In cancer, both tumor-suppressive and tumor-promoting functions have been described for autophagy depending on the transformation stage. In normal cells, autophagy functions as a tumor-suppressive mechanism by removing damaged proteins and organelles, maintaining low levels of ROS by the removal of damaged mitochondria, and removing proteins which promote pro-tumorigenic signaling, like p62/SQSTM1, or by inducing senescence after an initial oncogenic signal [10]. On the other hand, once transformation has occurred, tumor cells rely on autophagy to survive metabolic stress and hypoxia within the tumor, to maintain cancer stem cells (CSC) and to survive to anoikis and therapy [8]. Also, some types of cancer with certain oncogenic backgrounds have been shown to be dependent on autophagy, even in the absence of stress. This addiction to autophagy has been described for several tumors with driver mutations in the RAS/MAPK pathway and a similar dependence has been described for TNBC, in which autophagy has been shown to contribute to the maintenance of constitutively activated STAT3 [7]. Thus, several ongoing clinical trials are exploring autophagy inhibition using pharmacological agents in combination with cancer therapies in diverse types of cancers [11]. Chloroquine (CQ) and hydroxychloroquine (HCQ) are the only clinically available drugs to inhibit autophagy and both are currently being used for this purpose in clinical trials for treating cancer and other diseases without adverse toxicity [11]. Both CQ and HCQ are lysosomotropic amine-containing drugs, considered to decrease lysosomal acidification [12,13,14,15], and, more recently, CQ has been shown to block autophagosome–lysosome fusion without affecting lysosomal pH [16]. So, despite controversies regarding the precise cellular target, CQ is one of the most widely used drugs to block autophagy at the autolysosomal degradation step, known to induce the accumulation of autophagosomes due to their decreased turnover [16] and to decrease the degradation of autophagic substrates [17].

Recently, autophagy has been implicated in the regulation of protein secretion, either directly, in an unconventional protein secretion pathway termed secretory autophagy, or indirectly, by regulating the production of mitochondrial ROS which, in turn, regulate protein secretion, particularly the secretion of pro-inflammatory cytokines [18]. Regarding the latter, the pharmacological or genetic inhibition of autophagy has been shown to increase the secretion of IL-1β [19,20,21], IL-6 [22], IL-18 [19,20] and macrophage migration inhibitory factor (MIF) [23] in lipopolysaccharide (LPS)-treated macrophages and of IL-6 [22] in cancer epithelial cells. Importantly for cancer therapy, many of these cytokines have been shown to have pro-tumorigenic effects in cancer cells through the activation of pro-tumorigenic signaling pathways, CSC maintenance and the promotion of epithelial to mesenchymal transition (EMT) in different types of cancer [10]. In this work, we show that the inhibition of autophagy in TNBC cell lines promoted the secretion of MIF, a pro-inflammatory cytokine associated with the regulation of macrophage metabolism, phagocytosis, adherence, and T cell proliferation [24], and also associated with the promotion of malignancy, through the activation of MAPKs (Mitogen-Activated Protein Kinases), inhibition of p53 [25], induction of angiogenesis, metalloprotease secretion and immunomodulation through the activation of myeloid-derived suppressor cells [26,27,28].

## 2. Materials and Methods

### 2.1. Cell Lines and Cell Culture

EpH4-Ev (ATCC, CRL-3063) mouse epithelial breast cells were used as a non-tumorigenic control, mouse breast cancer cell lines used were: B-MEKDD 116 (ATCC, CRL-3069), which is a MEK1 transformed, tumorigenic cell line and cell lines from the 4T1 model of different metastatic capacities [29,30]: 67NR, non-metastatic; 168FARN, disseminates to lymph nodes; 66cl4, metastatic to lung; 4T1, metastatic to lung, liver, bone and brain. Lenti-X293FT (NC983960, Clontech, USA) cell line was used for lentivirus production. Cell lines were cultured in DMEM medium (DMP15, Caisson, UT, USA) supplemented with 10% Fetal Bovine Serum (FBS, S1560-500, Biowest, MO, USA) and 1% antibiotic (P5L01, Caisson). EpH4-Ev and B-MEKDD 116 cell lines were cultured in DMEM supplemented with 10% Fetal Calf Serum (S075R-500, Biowest, MO, USA) and 1% antibiotic. All cells were cultured in a humidified atmosphere of 95% air/5% CO2 at 37 °C. Proliferation and cell death analysis was performed in a real-time microscopy IncuCyte^®^ZOOM System (Essen BioScience, Ann Arbor, Michigan, USA). Proliferation data is presented as % confluence vs. time. Cell death was evaluated using 10 μM propidium iodide (PI, P4170, Sigma, MO, USA) staining for 10 min. Data for cell death are presented as percent confluency of red fluorescence (% confluency of PI+ cells) normalized to total cell confluency [31]. Cell migration experiments were performed using an Essen Bioscience WoundMaker^TM^ in confluent wells of a 96-well plate according to the manufacturer’s instructions. Cells were visualized every 4 h in an IncuCyte^®^ZOOM System. Wound closure results are expressed as wound closure area (area of the wound region covered by migrating cells as a function of the wound area at time 0) and were calculated using ImageJ software.

### 2.2. Reagents

Reagents used: Chloroquine diphosphate salt (CQ, C6628, Sigma, MO, USA), N-acetyl-L-cysteine (NAC, A7250, Sigma), ISO-1 (4288, TOCRIS, BS, UK), dihydroethidium (DHE, 37291, Sigma), Polyethylenimine (PEI, 23966, Polysciences, Inc., PA, USA).

### 2.3. Autophagy Gene Expression Plate

RNA was extracted using RNeasy Mini Kit (Qiagen, Hilden, Germany, 74104). A total of 1 μg of total RNA was used in an RT2 Profiler Mouse Autophagy PCR Array (Sabiosciences, Qiagen, Hilden, Germany, PAMM-084) according to the manufacturer’s instructions in a Roche LightCycler 480 (Roche Molecular Systems, Inc., Pleasanton, CA, USA). Expression levels were quantified relative to the values of the housekeeping genes. Data analysis was performed using Sabiosciences analysis software and expressed as differences in the magnitude of gene expression.

### 2.4. Lentiviral Production

Packaging plasmids (pVSV-G, pRRE, pRSV), as well as the plasmid containing the shRNA sequence (non-silencing (NS), ATG7 TRCN0000007587 or beclin-1 TRCN 0000087288 pLKO.1 vectors), were transfected into Lenti-X293FT cells using PEI (3 μg PEI/μg of DNA). Medium was changed 18 h after transfection and viruses were collected at 24 and 48 h and pooled. Polybrene (0.8 μL/mL) was added to the virus-containing media and was frozen until transduction. For transduction, breast cancer lines cells were plated at a density of 30,000–40,000 cells per well. Cells were incubated with polybrene (0.8 μL/mL) and 0.5–1 mL of virus-containing medium was added and incubated for 24 h. Transduced cells were selected with puromycin (Sigma, P8833): 67NR, 5 μg/mL; 66cl4, 3 μg/mL and 4T1, 2 μg/mL for 2–3 days.

### 2.5. Western Blot

Cells were trypsinized and lysed by sonication with RIPA buffer containing a protease inhibitor cocktail (Complete, 11697498001, Roche, MA, DE). Protein was quantified by Bradford assay. A total of 30 μg of protein were resolved by a 12% SDS-PAGE gel and transferred to polyvinylidene difluoride membranes (PVDF) (IPVH00010, Merck Millipore, Cork, IRL). Membranes were blocked with 5% milk in PBS-Tween 20 (Sigma, p1379) for 1 h at room temperature and then incubated overnight at 4 °C with the primary antibodies: anti-ATG7 (8558, 1:500, Cell signaling, MA, USA), anti-beclin-1 (3738, 1:1000, Cell signaling), anti-LC3 (NB100-2220, 1:1000, Novus Biologicals, CO, USA), anti-actin (A5441, 1:10,000, Sigma) or anti-VDP/P115 (PA5-30281, 1:1,000, ThermoFisher, IL, USA). Secondary antibodies used were HRP-linked anti-rabbit IgG (7074, 1:20,000, Cell signaling) and HRP-linked anti-mouse IgG (A2304, 1:20,000, Sigma). Immobilon™ Western (WBKLS0500, Millipore) detection reagents and a C-Digit Blot Scanner (LI-COR Biosciences, Lincoln, NE, USA) imaging system were used to visualize and digitalize the Western Blot images. Densitometric analysis was performed using Image Studio Lite Ver 5.3 software. The molecular weight was evaluated with pre-stained protein ladder, 10–180 kDa (26616, ThermoFisher, IL, USA).

### 2.6. Macrophage Migration Inhibitory Factor (MIF) Secretion

Secreted MIF was evaluated with a Duo Set mouse MIF ELISA kit (DY1978, R&D systems, MN, USA), according to the manufacturer’s protocol. Cells were plated at a density of 100,000 cells per well and, after 16 h, the supernatant was collected for MIF measurement and cells were trypsinized to quantitate total protein. Data are presented as MIF/total protein in the cell lysate (pg/pg) in order to normalize to cell number. For conditioned media (CM) experiments, 66cl4 cells were transduced with either a non-silencing (NS), ATG7 or beclin-1 (B1) shRNA, plated and, after 24 h, medium was replaced with fresh medium (+/−7 mM NAC) for CM collection. After 16 h, CM was collected and placed on confluent 67NR or 4T1 cells (+/−ISO-1) which had been scratch-wounded to evaluate cell migration with the wound closure experiment.

### 2.7. Pro-Inflammatory Cytokine Secretion

67NR, 66cl4 and 4T1 cells were plated at a density of 100,000 cells per well in 12-well-plates, and after 24 h the medium was replaced with fresh medium +/−CQ 10, 20 or 40 µM. After 16 h, the supernatant was collected for pro-inflammatory cytokine measurement using six bead populations coated with capture antibodies for Interleukin-6 (IL-6), Interleukin-10 (IL-10), Monocyte Chemoattractant Protein-1 (MCP-1), Interferon-γ (IFN-γ), Tumor Necrosis Factor (TNF) and Interleukin-12p70 (IL-12p70). Samples were prepared according to the Cytometric Bead Array (552364, BD, CA, USA), mouse inflammation kit instructions and analyzed in a BD FACS Canto II flow cytometer (CA, USA) using FCAP Array v3.0 software (BD).

### 2.8. ROS Evaluation

ROS levels were evaluated by flow cytometry and dihydroethidium (DHE) staining. 100,000 cells were plated in 12 well-plates and, after 16 h of treatment with CQ, stained with 10 µM DHE for 30 min at room temperature, protected from light, washed with PBS, trypsinized and centrifuged at 2500 rpm. Pellets were resuspended in PBS with 3% FBS and analyzed in a BD FACS Canto II flow cytometer. Results are expressed as percentage of DHE^high^ cells, which were considered as the cells in the 75th percentile. Treatment with 3.5 and 7 mM N-acetyl cysteine (NAC) was used in combination with CQ and, after 16 h, cells were collected and stained with DHE as previously described.

### 2.9. Statistical Analysis

Data show the mean +/− standard error of at least three independent experiments. One-way ANOVA was performed using GraphPad Prism 6.0c software (CA, USA). Post-hoc Tukey test was performed when means were compared to every other mean, and Dunnett’s post-hoc was used for multiple-to one comparison. For one-to-one comparisons, a student *t*-test was used using GraphPad Prism 6.0c software.

## 3. Results

### 3.1. Breast Cancer Cell Lines with Different Metastatic Capacities Differ in Basal Levels of Autophagy

We used breast cancer cell lines with different metastatic capacities. 67NR cells are tumorigenic but non-invasive, 168FARN cells disseminate to the lymph nodes only, 66cl4 cells are metastatic only to the lungs and 4T1 cells are highly metastatic and can disseminate to the lung, liver, bone and brain [29,30]. Hierarchical clustering of the expression level of *Atg* genes and genes related to autophagy revealed clustering according to the invasive capacity of the cell lines. Non-metastatic 67NR cells clustered together with the weakly invasive 168FARN cell line, then with 66cl4 cells (metastatic to lung), and finally with the highly metastatic 4T1 cell line (Figure 1A), indicating a relationship between the expression of genes involved in the autophagic pathway and the intrinsic metastatic ability, and also suggesting a possible association with levels of basal of autophagy and metastatic capacity. Basal autophagy was evaluated in metastatic cell lines and compared to the non-metastatic 67NR cells. Autophagic flux is often measured by LC3II turnover by Western blot. The measurement of LC3II using lysosomal inhibitors like chloroquine (CQ) to block autophagosome degradation, can be used as an indication of the amount of autophagosomes present in a certain condition. A comparison of the accumulation of LC3II in the presence of a lysosomal inhibitor between different cell lines can be used as a measurement of basal levels of autophagy [32]. CQ treatment in breast cancer cell lines induced LC3II accumulation and densitometric analysis showed higher LC3II accumulation in the metastatic cell lines (66cl4 and 4T1) when compared to the non-metastatic one (67NR, Figure 1B). Also, metastatic cells were more sensitive to CQ treatment than the non-metastatic cell line. CQ treatment decreased cell viability (Appendix A) and increased cell death (Figure 1C) more in the metastatic (66cl4 and 4T1) than in the non-metastatic (67NR) cell lines. Importantly, basal autophagy levels were not directly related to higher metastatic ability, since the 66cl4 cell line, which only metastasizes to the lung, had the highest levels of basal autophagy (Figure 1B) and was the most sensitive to CQ treatment (Figure 1C and Appendix A).

### 3.2. Inhibition of Autophagy Induced the Secretion of Macrophage Migration Inhibitory Factor (MIF) in Breast Cancer Cell Lines

MIF has been related to several aspects involved in the progression of malignancy [27,33]. To test a possible relationship between MIF secretion and malignancy in a panel of breast cancer cell lines, we evaluated the basal secretion of MIF to the culture media using the EpH4-Ev mouse epithelial breast cells as a non-tumorigenic control; B-MEKDD 116, which is a MEK1 transformed and tumorigenic cell line; 67NR, non-metastatic; 66cl4, metastatic to the lung, and 4T1, highly metastatic cell lines. We did not find a relationship between the basal secretion of MIF with the invasive phenotype in the mouse breast cancer cell lines studied and found increased levels of basal MIF secretion in the 67NR non-metastatic cells when compared to the non-tumorigenic control and to 4T1 cells (Figure 2A).

The pharmacological inhibition of autophagy with CQ treatment induced MIF secretion in all the cell lines tested at 16 h of treatment (Figure 2B) and induced the highest secretion in the 66cl4 cells, where an increase in extracellular MIF was detected with 20 or 40 µM CQ. Only the highest concentration of CQ evaluated (40 µM) induced the secretion of MIF in the 67NR and the 4T1 cell lines. Importantly, MIF secretion in the three cell lines occurred without a significant induction of cell death at this timepoint (Appendix A), indicating that the secretion of MIF was not related to loss of plasma membrane integrity occurring during cell death, since no increase in PI staining at 16 h was observed after CQ treatment. These data indicate that breast cancer cell lines secreted MIF to the culture media in response to CQ treatment before cell death could be observed.

In order to discard possible autophagy-independent effects of CQ treatment and to test if the effect on MIF secretion was particular to the lysosomal inhibition of autophagy, we used shRNAs to block autophagy at early steps of the pathway with ATG7 or beclin-1 (B1) shRNAs. ATG7 knockdown induced MIF secretion in the 66cl4 cell line, and beclin-1 knockdown increased MIF secretion to the culture media in both the 67NR and 66cl4 cell lines (Figure 2C,D). Protein levels after shRNA viral transduction and puromycin selection were evaluated by Western blot (Figure 2C,D). ATG7 knockdown efficiency was similar in the three cell lines tested (Figure 2C) and beclin-1 knockdown was more efficient in the 66cl4 cells than in the 67NR or the 4T1 cell lines (Figure 2D). In order to test that beclin-1 silencing was effective in all cell lines, knockdown efficiency in the 67NR or 4T1 cell lines was also evaluated at a higher viral titer, where a better knockdown efficiency was achieved (Appendix A). However, since MIF secretion was supposed to be evaluated in the absence of cell death, and since the 66cl4 cell line was the one with the highest sensitivity to autophagy inhibition and highest MIF secretion in response to CQ or ATG protein knockdown, we used the viral titer shown in Figure 2C,D for further knockdown experiments in the 66cl4 cell line.

Other pro-inflammatory cytokines (MCP1 and IL-6) were released to the culture media at the same time in the 66cl4 cell line (Figure 2E,F) but not in the 67NR or 4T1 cells (Appendix A) after CQ treatment. MCP1 secretion increased with 10 and 20 µM CQ treatment and an increase in IL-6 was detected only with 20 µM CQ, probably due to different secretion kinetics and to a faster induction of cell death at higher concentrations of CQ. Besides MCP1 and IL-6, other pro-inflammatory cytokines were evaluated (IL-10, IFN-γ, TNF and IL-12p70) but were not detected in the supernatants, at least not within the dynamic range evaluated. Our data indicate an important role for the secretion of pro-inflammatory cytokines en route to cell death induced by the inhibition of autophagy, and that the inhibition of the autophagic process (pharmacologic or genetic) induces the secretion of MIF, particularly in the 66cl4 breast cancer cell line.

### 3.3. MIF Secretion Induced by the Inhibition of Autophagy was Mediated by Reactive Oxygen Species (ROS) Production and Autocrinally Induced Cell Survival

We evaluated ROS production in response to the inhibition of autophagy and found increased levels of ROS after CQ treatment in the 66cl4 and 4T1 cell lines (Figure 3A). Also, CQ treatment increased Uso1/p115 protein levels (Figure 3B) at early timepoints (2 or 4 h) and decreased in the 66cl4 and 4T1 cell lines at 8 h (Figure 3B), probably suggesting p115 secretion with MIF in the latter cell lines and a differential involvement of ROS and p115 in the three different cell lines studied. Importantly, ROS inhibition with N-acetylcysteine (NAC) treatment in the 66cl4 cell line (Figure 3C) decreased the secretion of MIF to the culture medium (Figure 3C), and treatment with ISO-1, a MIF inhibitor [24], increased cell death in the 66cl4 cell line treated with CQ (Figure 3D,E) but not in the 67NR or 4T1 cell lines (Appendix A), indicating that MIF secretion induced by the inhibition of autophagy in the 66cl4 cell line has an important autocrine effect on cell survival.

### 3.4. MIF Secretion Induced by the Inhibition of Autophagy Induced Migration in Autophagy-Proficient Breast Cancer Cell Lines

Since cells of different metastatic abilities are found in a tumor, and since the three cell lines we used for this study were isolated from the same, spontaneously arising tumor in a mouse, we evaluated the paracrine effect of secretion induced by the inhibition of autophagy in other cell lines where autophagy was not being inhibited. We inhibited autophagy using shRNAs (NS, non-silencing control, ATG7 or beclin-1 shRNAs) in the 66cl4 breast cancer cell line, which showed the highest level of pro-inflammatory cytokine secretion in response to the inhibition of autophagy (Figure 2), and in which ROS production was found to mediate MIF secretion (Figure 3); collected conditioned media (CM) from these cells and incubated non-transduced, autophagy proficient 67NR or 4T1 cells with this CM. Incubation with the CM containing the secreted factors from the autophagy-deficient (ATG7 or beclin-1 shRNA-expressing) 66cl4 cells increased the migration efficiency of the 67NR and 4T1 cell lines (Figure 4A–C and Appendix A). Importantly, this increased migration was prevented by ISO-1 (added directly to CM) or by NAC treatment (added to 66cl4 cells during CM collection, to avoid ROS-mediated secretion) (Figure 4C and Appendix A), indicating that autophagy inhibition induced a ROS-mediated secretion of MIF in the 66cl4 cells, which was implicated in the paracrine promotion of malignancy in other breast cancer cells.

## 4. Discussion

Macrophage migration inhibitory factor (MIF) is a pleiotropic, pro-inflammatory mediator with an important role in the innate and adaptive immune system [26,34]. It is known to be produced and secreted by numerous immune cells [34] and is also expressed in many endocrine, epithelial and endothelial cells [26]. In contrast to most cytokines, it is constitutively expressed, stored in intracellular vesicles and, since it does not have an endoplasmic reticulum (ER)-localizing sequence to enter the classical ER–Golgi complex pathway for secretion [35], it is secreted by an unconventional protein secretion pathway, known to involve ABC transporters [36], P115 [37] and the inhibition of autophagy [23] in LPS-activated macrophages. MIF has been implicated in several autoimmune and inflammatory disorders and, recently, it has been related to cancer progression due to the signaling pathways it activates [26]. Extracellular MIF can bind to its receptors on the cell surface. When binding CD74/CD44, it activates the MAPK/ERK pathway and can trigger an immune response through autocrine and paracrine signaling via the induction of pro-inflammatory cytokines such as TNF-∝, IL-1β, IL-6, nitric oxide, COX2 and IFN-γ [26]. MIF can also bind chemokine receptors CXCR2/4, facilitating chemotactic responses through the induction of IL-8 and recruiting monocytes, lymphocytes and neutrophils to the sites of inflammation [26]. Importantly, signaling pathways activated by MIF and the pro-inflammatory cytokines, secreted after MIF binding to their receptors, have been shown to have important pro-tumorigenic effects in breast and other types of cancer [10,38]. In breast cancer, several studies have implicated MIF in the promotion of malignancy: elevated MIF levels have been found in the tissue sections [39,40,41] and sera of breast cancer patients when compared to healthy individuals [40]; MIF has been shown to have an important role in the promotion of cellular proliferation and invasion in vitro [39]; and it has also been shown to have an anti-tumoral immunosuppressive role in vivo [28]. The above-mentioned evidence indicates that MIF has a pro-tumorigenic role in breast cancer and that it should be considered as a target for breast cancer therapy.

In cancer, autophagy has been related to several aspects of the invasive process, like promoting survival to anoikis [42], regulating EMT [10], and promoting tumor cell migration and invasion [10] and it has been suggested as a promising target for therapy, particularly in those types of cancer that are dependent on autophagy for survival [11] as has been previously demonstrated for triple negative breast cancer (TNBC) [7]. In this work, we found that the inhibition of autophagy in TNBC cell lines of different metastatic abilities induced cell death but also induced the secretion of MIF before the loss of plasma membrane integrity could be detected (Figure 2B, Appendix A). Basal MIF secretion was not related to the invasive capacity of the different cell lines tested (Figure 2A) and the secretion of MIF mediated by the inhibition of autophagy (pharmacologic or genetic) was the highest in the 66cl4 cell line when compared to the other cell lines tested (Figure 2). Also, MCP1 and IL-6 secretion were induced in the 66cl4 and not in the 4T1 or 67NR cell lines by chloroquine (CQ) treatment, suggesting an important pro-inflammatory cytokine-secreting role in response to the inhibition of autophagy in the 66cl4 cell line.

Basal autophagy is defined as the constitutive autophagic degradation of cytoplasmic material that proceeds in the absence of any overt stress or stimulus [32]. We found higher levels of basal autophagy in the metastatic cell lines when compared to the non-metastatic one, indicating differences in the intrinsic regulation of autophagy that could promote their invasive capacities or that could render them less sensitive to stress occurring during the metastatic process. However, basal levels of autophagy were not directly related to the invasive capacity of the cells, since the 4T1 cell line, with the highest metastatic ability, was not the one with the highest levels of basal autophagy (Figure 1). Conversely, the 66cl4 cell line, which only metastasizes to the lung, had the highest levels of basal autophagy. Our data show that the basal levels of autophagy correlated with the sensitivity to CQ treatment (Figure 1) and higher pro-inflammatory cytokine secretion in response to the inhibition of autophagy (Figure 2). One of the pro-inflammatory cytokines, which was secreted by the inhibition of autophagy, particularly in the 66cl4 cell line, was MIF. To our knowledge, this is the first evidence that the inhibition of autophagy induces the secretion of MIF in cancer cells. Since CQ is known to have effects independent of its role in blocking autolysosomal degradation in autophagy [43], and since inhibiting the degradation step of autophagy could be different to inhibiting autophagosome formation, we used the genetic inhibition of autophagy-targeting autophagosome elongation (with ATG7 shRNA) or nucleation (with beclin-1 shRNA). The genetic inhibition of autophagy also increased MIF secretion to the culture media in the 66cl4 cell line, indicating that blocking the autophagic process in cancer cells induced the secretion of MIF and that the effect was not limited to pharmacological or lysosomal inhibition of autophagy.

One of the consequences of blocking the autophagic pathway is an increase in ROS, due to the accumulation of damaged mitochondria [23]. MIF secretion in the 66cl4 cell line was due to increased ROS production caused by the inhibition of autophagy, since it could be prevented by treatment with an antioxidant (Figure 3C). Secreted MIF had important autocrine effects in the promotion of malignancy, since CQ-treated 66cl4 cells dramatically increased cell death when co-treated with ISO-1, a MIF inhibitor (Figure 3D,E). Also, importantly for the promotion of malignancy, conditional media (CM) from autophagy-deficient 66cl4 cells increased the migration efficiency in the 67NR and 4T1 cell lines, indicating a paracrine effect of cytokine secretion in the promotion of malignancy (Figure 4). Increased migration could be decreased by treatment with an antioxidant (NAC) during media collection of autophagy-deficient (ATG7 shRNA-expressing) 66cl4 cells, indicating an important role of reactive oxygen species (ROS) production, mediated by the inhibition of autophagy in the secretion of pro-tumorigenic factors, and was also prevented by ISO-1’s addition to the collected CM to block MIF-related effects, indicating an important role for secreted MIF mediated by the inhibition of autophagy in the paracrine promotion of malignancy.

In the 4T1 model, cell lines with different metastatic capacities were isolated from a single, spontaneously arising mammary tumor from a Balb/cfC_3_H mouse [29]. Our data indicate that in a tumor in which cells with different metastatic abilities and different levels of basal autophagy are likely to be found, treatment with autophagy inhibitors is likely to induce cell death with a concomitant increase in pro-inflammatory cytokine production by tumor cells. We have shown that, among these secreted factors, MIF will have an important pro-tumorigenic role, as it had an autocrine tumor-promoting effect by inducing cell survival and a paracrine effect in neighboring cells by inducing tumor cell migration. Since MIF has been shown to have a tumor-immunomodulatory role by inducing myeloid derived suppressor cell (MDSC) accumulation within the tumor [28], it is likely that the inhibition of autophagy will also induce immunomodulation in breast cancer tumors, but these effects remain to be tested.

## 5. Conclusions

Several studies have implicated autophagy in protein secretion, particularly for proinflammatory cytokines. However, few studies have addressed the role of autophagy in proinflammatory cytokine secretion in the context of cancer therapy. In this work, we show that, for triple negative breast cancer (TNBC), an autophagy-dependent type of cancer, breast cancer cells treated with chloroquine (CQ), one of the drugs most commonly used to inhibit autophagy in cancer clinical trials, or with the genetic inhibition of autophagy, increased reactive oxygen species (ROS) production, probably due to damaged mitochondria accumulation, and secreted macrophage migration inhibitory factor (MIF) to the culture media. Importantly, secreted MIF had a pro-survival effect in the cell line with the highest levels of MIF secretion in response to autophagy inhibition and paracrinally induced cell migration in the other two cell lines. Our results strongly suggest that the inhibition of autophagy should be combined with pro-inflammatory cytokine modulation, and particularly MIF inhibitors, for a better outcome during TNBC cancer therapy.

## Figures and Tables

**Figure 1 biology-09-00020-f001:**
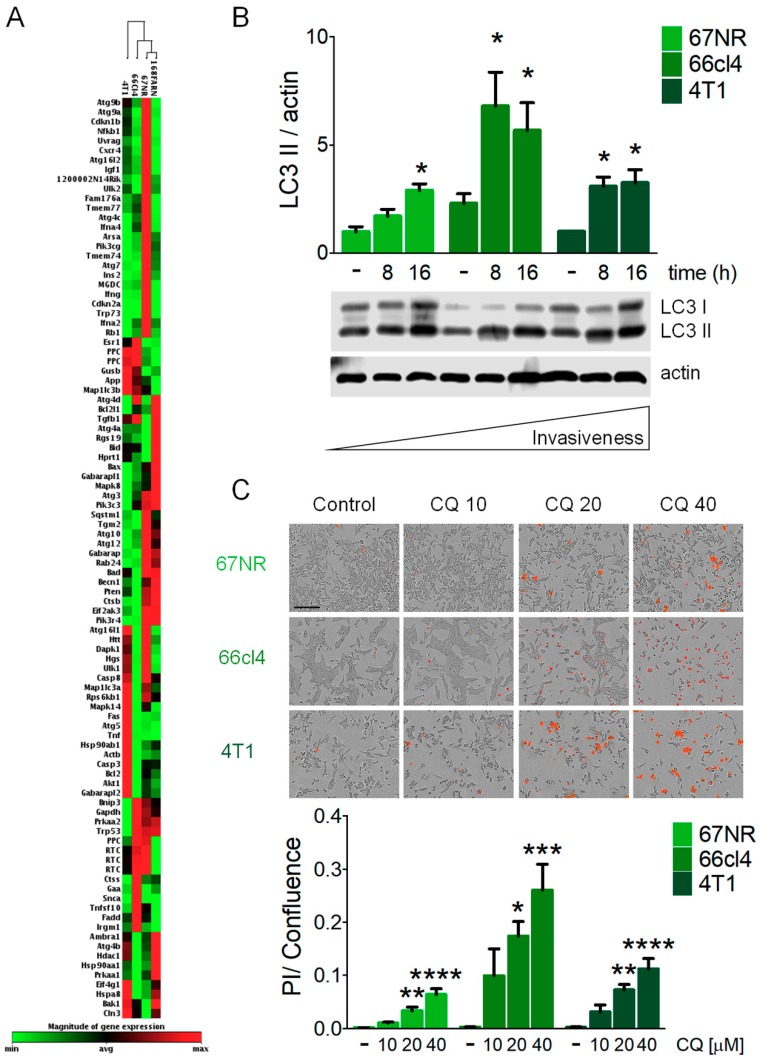
**Triple Negative Breast Cancer** (TNBC) cell lines with different metastatic capacities differ in basal levels of autophagy and sensitivity to autophagy inhibition. (**A**) Hierarchical clustering analysis of autophagy-related gene expression clustered cell lines according to their metastatic capacity (67NR, non-metastatic; 168FARN, weakly metastatic; 66cl4, metastatic only to lung; 4T1, highly metastatic). (**B**) LC3II accumulation was evaluated by Western blot for basal autophagy assessment using 10 µM chloroquine (CQ) at the indicated times. (**C**) Cell death evaluation with propidium iodide (PI) staining was assessed after 24 h of treatment with the indicated concentrations of CQ [µM]. The scale bar in the pictures in (**C**) represents 200 µm. Graphs shows mean +/− standard error of three to four independent experiments, * *p* < 0.05, ** *p* < 0.01, *** *p* < 0.001, **** *p* < 0.0001.

**Figure 2 biology-09-00020-f002:**
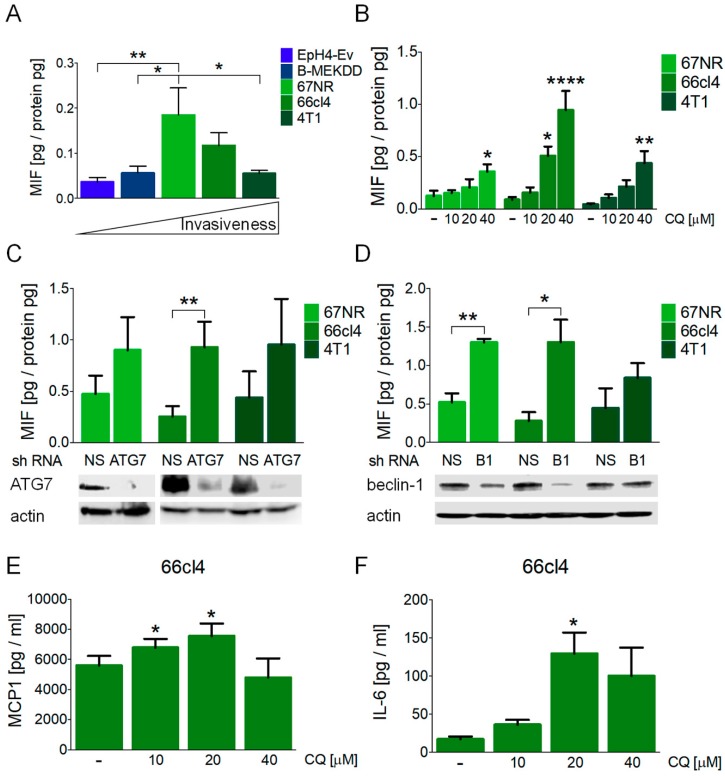
Autophagy inhibition induced the secretion of macrophage migration inhibitory factor (MIF) in breast cancer cell lines. (**A**) Basal secretion of MIF to the culture media was evaluated in the EpH4-Ev (mouse epithelial breast cells), B-MEKDD 116 (MEK1 transformed, tumorigenic cell line), 67NR (non-metastatic), 66cl4 (metastatic to lung) and 4T1 (highly metastatic) cell lines. Media was collected at 16 h. Autophagy inhibition with CQ treatment (**B**), ATG7 knockdown (**C**) or beclin-1 (B1) knockdown (**D**), all collected after 16 h, induced the secretion of MIF in the breast cancer cell lines studied. In (**C**,**D**), ATG7 or beclin-1 protein levels were evaluated by Western blot to confirm knockdown efficiency and are shown below the graphs. Autophagy inhibition with CQ treatment also induced the secretion of MCP-1 (**E**) and IL-6 (**F**) in the 66cl4 breast cancer cell line at 16 h. Graphs show mean +/− standard error of three-four independent experiments, * *p* < 0.05, ** *p* < 0.01, **** *p* < 0.0001. NS, non-silencing shRNA; B1, beclin-1.

**Figure 3 biology-09-00020-f003:**
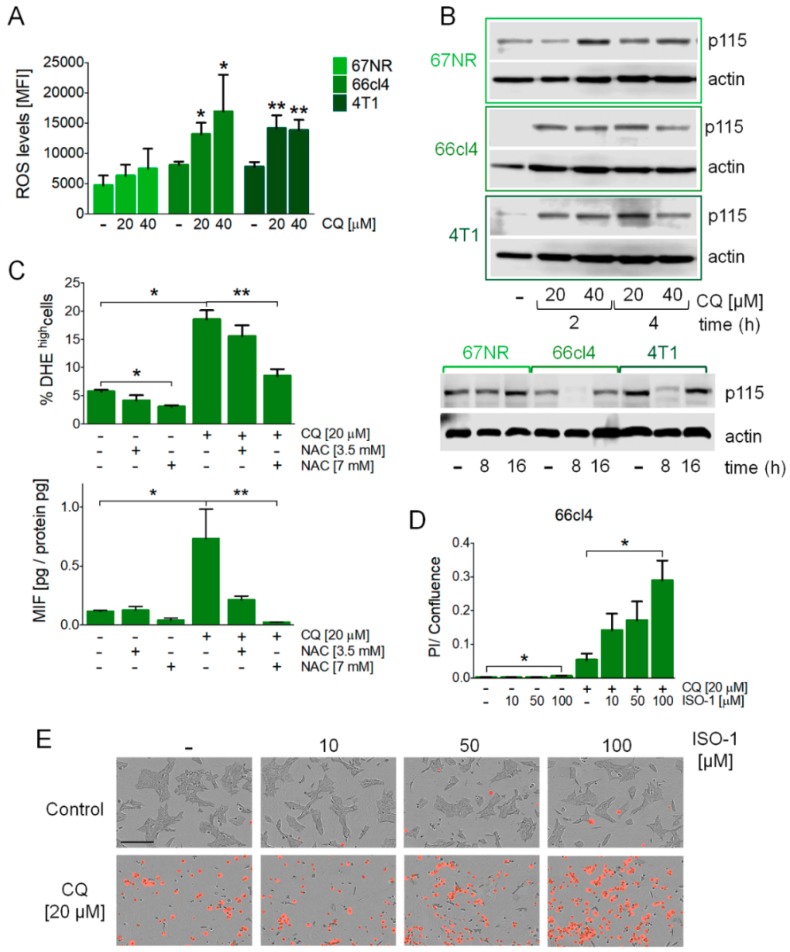
MIF secretion induced by the inhibition of autophagy was mediated by reactive oxygen species (ROS) production, occurred with changes in Uso1/p115 protein levels, and mediated cell survival in the 66cl4 cell line. ROS levels were evaluated by DHE staining by flow cytometry at 16 h of treatment at the indicated concentrations of CQ (**A**). Uso1/p115 protein levels were evaluated by Western blot in the three cell lines studied (**B**). N-acetyl cysteine (NAC) treatment at the indicated concentrations reduced the ROS levels induced by CQ treatment and reduced MIF secretion in the 66cl4 cell line (**C**) both at 16 h of treatment. ISO-1, a MIF inhibitor, increased cell death in the 66cl4 cell line when used in combination with CQ treatment for 24 h (**D**,**E**). The scale bar in the pictures in (**E**) represents 200 µm. Results show mean +/− standard error of three-four independent experiments, * *p* < 0.05, ** *p* < 0.01.

**Figure 4 biology-09-00020-f004:**
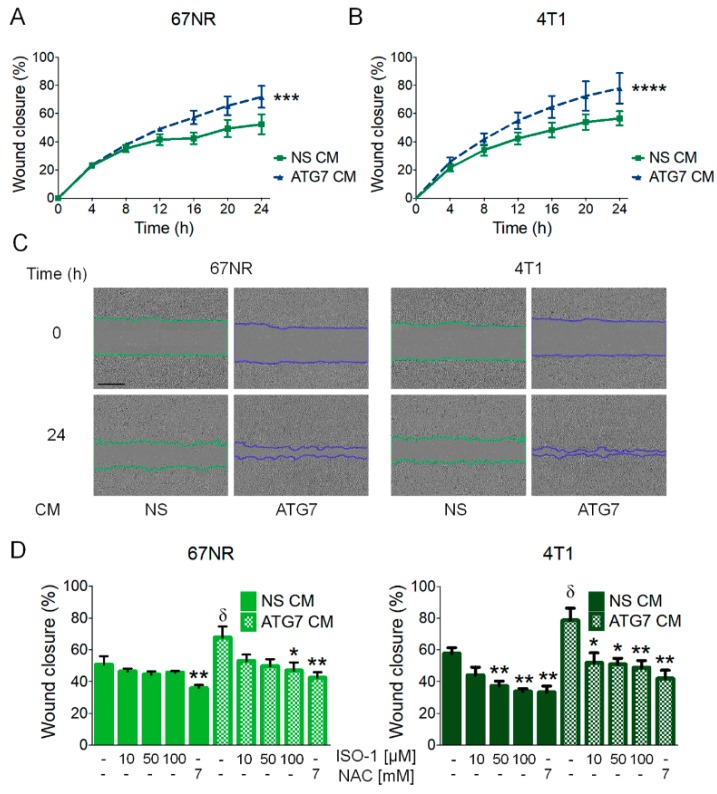
MIF secretion induced by the inhibition of autophagy paracrinally induced cell migration in autophagy-proficient breast cancer cell lines. Conditioned medium (CM) collected from 66cl4 cells transduced with a non-silencing (NS CM) or ATG7 (ATG7 CM) shRNA, was added to wild type 67NR or 4T1 cells for migration evaluation in a scratch-wound assay (**A**,**B**). Representative images of 67NR or 4T1 cells treated with 66cl4 CM are shown; the scale bar in the pictures represents 400 µm (**C**). Increased migration in 67NR or 4T1 cells induced by ATG7 CM from 66cl4 cells was decreased when 66cl4 ATG7 shRNA-expressing cells were treated with an antioxidant (NAC) during CM collection or when an MIF inhibitor (ISO-1) was added to the CM at the indicated concentrations (**D**). Graphs show wound closure area covered by the migrating cells at the indicated times or at 24 h. Graphs show mean +/− standard error of three to four independent experiments, * *p* < 0.05, ** *p* < 0.01, *** *p* < 0.001, *****p* < 0.0001. In (**D**), δ: different from NS CM untreated control, *: different from their respective untreated control.

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
