# Peer review of "Autophagy Inhibition Induces the Secretion of Macrophage Migration Inhibitory Factor (MIF) with Autocrine and Paracrine Effects on the Promotion of Malignancy in Breast Cancer"

_biology, 2020, doi:10.3390/biology9010020_

Round 1

Reviewer 1 Report

The manuscript by Cotzomi-Ortega et al., tested the effect of chloroquine on multiple breast cancer cell lines and found that chloroquine induced the secretion of macrophage migration inhibitor factor (MIF) which affected breast cancer cells invasion. Although the work is interesting, however the major concern about the conclusion made by authors is that they assumed all the effect of chloroquine is mediated through induction of autophagy, but the authors do not have a strong experimental data to support their claim. A significant work is required to demonstrate whether the effect of chloroquine is mediated through autophagy. Moreover, the authors need to test other autophagy inducing factors such as metabolic deprivation/serum starvation in expression of MIF. Also, induction of LC3I/LC3II is not sufficient to conclude autophagy flux.  Additional note: some of the graphs presented do not match the blots shown.

Author Response

We would like to thank the reviewers for their careful revision of our manuscript. We consider your comments have greatly improved the quality of our paper. As a response to both reviewers, we double checked English language and style throughout the manuscript and corrected some grammar and typo errors that were present in the text.

Below is our point-by-point response with our comments in red. All changes have been tracked on the revised manuscript.

Reviewer 1

The manuscript by Cotzomi-Ortega et al., tested the effect of chloroquine on multiple breast cancer cell lines and found that chloroquine induced the secretion of macrophage migration inhibitor factor (MIF) which affected breast cancer cells invasion.

Major comment:
Although the work is interesting, however the major concern about the conclusion made by authors is that they assumed all the effect of chloroquine is mediated through induction of autophagy, but the authors do not have a strong experimental data to support their claim.

Thank you for your comment. We do not assume that the effect of chloroquine is mediated by the induction of autophagy. In fact, CQ is a pharmacological inhibitor of autophagy. We apologize if we did not make this clear in the text. In order to clarify, we added the following text in the introduction and cite the Guidelines for the use and interpretation of assays for monitoring autophagy 3rd edition, (Klionsky D.J. et al, Autophagy, 2016):

Chloroquine (CQ) and hydroxychloroquine (HCQ) are the only clinically available drugs to inhibit autophagy and both are currently being used for this purpose in clinical trials for treating cancer and other diseases without adverse toxicity [11]. Both drugs accumulate in the lysosomes, decrease lysosomal acidification and block the degradation of the autophagosomes, inhibiting autophagy at the autolysosomal degradation step [12].

A significant work is required to demonstrate whether the effect of chloroquine is mediated through autophagy.

Thank you for your comment. We agree with the reviewer that a phenotype observed after treatment with chloroquine needs to be confirmed by additional experiments in order to prove that the effect is mediated by the inhibition of autophagy. In order to test this, we performed experiments using genetic inhibition of autophagy with shRNAs to knockdown ATG proteins and, in order to rule out effects specific to a particular ATG protein, we knocked down two different ATG proteins, as recommended in the current Guidelines for the use and interpretation of assays for monitoring autophagy (Klionsky D.J. et al, Autophagy, 2016).  In figure 2, we demonstrate that the secretion of MIF is induced by CQ treatment and by ATG7 or beclin-1 knockdown, at least in the 66cl4 cell line which is the one that we use for the migration experiments. All the migration experiments performed with conditional medium from autophagy deficient 66cl4 cells were performed using genetic inhibition of autophagy with ATG7 shRNAS. Moreover, in this revised version of the manuscript, we added data with beclin-1 shRNA (Supplementary Figure 3) further supporting our conclusions. Thus, we consider that our data strongly supports the main idea in our manuscript which is that the inhibition of autophagy induces the secretion of MIF and that secreted MIF influences malignancy.

In order to make this clear in the text, we added the following text in the conclusion:

Since CQ is known to have cytotoxic effects independent of its role in blocking autolysosomal degradation in autophagy [37], and since inhibiting the degradation step of autophagy could be different from inhibiting autophagosome formation, we used genetic inhibition of autophagy targeting autophagosome elongation (with ATG7 shRNA) or nucleation (with beclin-1 shRNA). Genetic inhibition of autophagy also increased MIF secretion to the culture media in the 66cl4 cell line, indicating that blocking the whole autophagic process in cancer cells induced the secretion of MIF and that the effect is not limited to pharmacological or lysosomal inhibition of autophagy.

Moreover, the authors need to test other autophagy inducing factors such as metabolic deprivation/serum starvation in expression of MIF.

Thank you for your comment. As we explained in the previous responses to the reviewer, the aim of this work was not to induce autophagy but to block autophagy. Indeed, it would be interesting to evaluate autophagy induction with regards to its effects on protein secretion. However, since the main approach that is being used in cancer clinical trials involves the inhibition of autophagy using lysosomal inhibitors like CQ or HCQ, with this manuscript we want to underscore possible undesirable effects that could happen during cancer therapy with autophagy inhibitors, like pro-inflammatory or MIF secretion. We thus write on conclusion:

Our results strongly suggest that inhibition of autophagy should be combined with pro-inflammatory cytokine modulation and particularly MIF inhibitors for a better outcome during TNBC cancer therapy.

Also, induction of LC3I/LC3II is not sufficient to conclude autophagy flux. 

Thank you for your comment. We agree with the reviewer that the induction of LC3I/LC3II is not sufficient to make conclusions about autophagic flux. The method we used to asses basal autophagic flux was LC3II turnover by Western Blot as suggested in the current Guidelines for the use and interpretarion of assays for monitoring autophagy (Klionsky D.J. et al, Autophagy, 2016). We thus used CQ to block autophagic flux and measured LC3II accumulation.  A comparison on the accumulation of LC3II in the presence of a lysosomal inhibitor between different cell lines can be used as a measurement of basal levels of autophagy. The results are presented in Figure 1B where LC3II was normalized to actin, again, as suggested in Klionsky D.J. et al, Autophagy, 2016.

Some of the graphs presented do not match the blots shown.

Thank you for your comment. We double checked all our graphs and blots and made sure that all the graphs presented match the blots shown. We think the reviewer refers to Figures 2C and D. However, the graph represents MIF secretion and the blots below represent knockdown efficiency. In order to clarify this, we added the following to the figure legend:

In (C) and (D), ATG7 or beclin-1 (B1) protein levels were evaluated by Western Blot to confirm knockdown efficiency and are shown below the graphs; NS, non-silencing shRNA.

We also changed the results description for figure 2 as follows in order to explain this:

ATG7 knockdown induced MIF secretion in the 66cl4 cell line and beclin-1 knockdown increased MIF secretion to the culture media in both the 67NR and 66cl4 cell lines (Figure 2 C,D). Protein levels after shRNA viral transduction and puromycin selection were evaluated by Western Blot (Figure 2 C,D). ATG7 knockdown efficiency was similar in the three cell lines tested (Figure 2C) and beclin-1 knockdown was better in the 66cl4 cells than in the 67NR or the 4T1 cell lines (Figure 2D).

Reviewer 2 Report

Authors of this manuscript reported a study that autophagy inhibition by small-molecule compounds induces migration inhibitory factor (MIF) secretion in the mouse triple negative breast cancer (TNBC) cell lines. The secreted MIF promotes the aggressiveness of TNBC cell lines. The study will provide information to therapeutic development targeting TNBC in human and is particularly relevant to the treatment strategy of combining autophagy inhibitor and other therapeutics. The study is well-designed. I have a few comments for the authors to address.

Knocked-down efficiency of beclin-1 in 4T1 is low based on the western blot. Is the statement for 4T1 robust? In Fig. 2, CQ at 40 uM decreases MCP1 and IL-6 secretion relative to 20 20 uM. Can the authors provide the explanation? In Fig. 3B, p115 expression in 66cl4 at 8h appears to have a different exposure time. Can the author confirm and explain? Authors described a panel of 8 cytokines were included in the detection in the Materials and Methods section. Only MCP1 and IL6, MIF were discussed. What are the secretion profile of other cytokines?

Minor point.

What does “bar in ( C ) represents 200 μm” in the Fig. 1 caption mean?

Author Response

We would like to thank the reviewers for their careful revision of our manuscript. We consider your comments have greatly improved the quality of our paper. As a response to both reviewers, we double checked English language and style throughout the manuscript and corrected some grammar and typo errors that were present in the text.

Below is our point-by-point response with our comments in red. All changes have been tracked on the revised manuscript.

Reviewer 2

Authors of this manuscript reported a study that autophagy inhibition by small-molecule compounds induces migration inhibitory factor (MIF) secretion in the mouse triple negative breast cancer (TNBC) cell lines. The secreted MIF promotes the aggressiveness of TNBC cell lines. The study will provide information to therapeutic development targeting TNBC in human and is particularly relevant to the treatment strategy of combining autophagy inhibitor and other therapeutics. The study is well-designed.

Minor comment:

Knocked-down efficiency of beclin-1 in 4T1 is low based on the western blot. Is the statement for 4T1 robust?

Thank you for your comment. We agree with the reviewer that knockdown of beclin-1 for the 4T1 cells is not robust in figure 2 and we had not explained this previously in the text. We know the shRNAs work fine for all the cell lines and in order to prove this we included in the revised version Supplementary Figure 1E, in which beclin-1 silencing is efficient for 67NR or 4T1 cells. However, since we wanted to evaluate MIF secretion in the absence of cell death (since loss of plasma membrane integrity would result in cytoplasmic content release to the culture media and increased MIF in the supernatants), we used a viral titer in which the 66cl4 cell line (which is the only one we used for subsequent experiments) showed an efficient knockdown. We changed the results description for figure 2 as follows in order to explain this:

In order to discard possible autophagy-independent effects of CQ treatment and to test if the effect on MIF secretion was particular to the lysosomal inhibition of autophagy, we used shRNAs to block autophagy at early steps of the pathway with ATG7 or beclin-1 shRNAs. ATG7 knockdown induced MIF secretion in the 66cl4 cell line and beclin-1 knockdown increased MIF secretion to the culture media in both the 67NR and 66cl4 cell lines (Figure 2 C,D). Protein levels after shRNA viral transduction and puromycin selection were evaluated by Western Blot (Figure 2 C,D). ATG7 knockdown efficiency was similar in the three cell lines tested (Figure 2C) and beclin-1 knockdown was better in the 66cl4 cells than in the 67NR or the 4T1 cell lines (Figure 2D). In order to test that beclin-1 silencing was effective in all cell lines, knockdown efficiency in the 67NR or 4T1 cell lines was also evaluated at a higher viral titer where a better knockdown efficiency was achieved (Supplementary Figure 1E). However, since MIF secretion was supposed to be evaluated in the absence of cell death and since the 66cl4 cell line was the one with the highest sensitivity to autophagy inhibition and highest MIF secretion in response to CQ or ATG protein knockdown, we used the viral titer shown in Figure 2C,D for further knockdown experiments in the 66cl4 cell line.

In Fig. 2, CQ at 40 mM decreases MCP1 and IL-6 secretion relative to 20 mM. Can the authors provide the explanation?

Thank you for your comment. We propose that differences in pro-inflammatory cytokine secretion are due to a different sensitivities to the inhibition of autophagy. Our data supports a main pro-inflammatory cytokine secretion of the 66cl4 cell line, which had the highest sensitivity to CQ treatment and the highest levels of basal autophagy. Thus, we could possibly explain changes in MCP1 and IL-6 secretion by a faster induction of cell death with 40 mM CQ treatment. It is thus possible that 40 mM CQ treatment induced secretion at earlier time points that were not detected with our experiment. We explain this in the text as follows:

Other pro-inflammatory cytokines (MCP1 and IL-6) were released to the culture media at the same time in the 66cl4 cell line (Figure 2E,F) but not in the 67NR or 4T1 cells (Supplementary Figure 2A,B) after CQ treatment. MCP1 secretion increased with 10 and 20 mM CQ treatment and an increase in IL-6 was detected only with 20 mM CQ, probably due to different secretion kinetics and to a faster induction of cell death at higher concentrations of CQ.

In Figure 3B, p115 expression in 66cl4 at 8h appears to have a different exposure time. Can the author confirm and explain?

Thank you for your comment. We apologize for this confusion. The membrane in Figure 2B is the same for all the cell lines so it does not have a different exposure time. We removed the lines between the bands in order to make this clear. As we explain in the text, we think this reflects p115 secretion together with MIF in this cell line, as has been previously shown for macrophages, but this hypothesis remains to be tested.

Authors described a panel of 8 cytokines were included in the detection in the Materials and Methods section. Only MCP1 and IL6, MIF were discussed. What are the secretion profiles of other cytokines?

Thank you for your comment. We only show results for MCP1 and IL-6 because the other cytokines were under the detection limit of the dynamic range evaluated: IL-10, 17.5 (pg/ml); IFNg, 2.5 (pg/ml); TNF, 7.3 (pg/ml); IL-12p70, 10.7 (pg/ml); for the three cell lines tested.

In order to clarify, we included the following in the results section:

Besides MCP1 and IL-6, other pro-inflammatory cytokines were evaluated (IL-10, IFN-g, TNF and IL-12p70) but were not detected in the supernatants, at least not within the dynamic range evaluated.

What does “bar in ( C ) represents 200 μm” in the Fig. 1 caption mean?

Thank you for your comment and we apologize for the lack of clarity. Bar in (C) represents the scale bar of the pictures in the figures. We changed the statement for “Scale bar represents 200 or 400 mm in Figure 1, 2 and 4 captions.

Round 2

Reviewer 1 Report

Overall, the revised manuscript has significantly improved. The authors also have addressed most of my previous concerns. I have not further comments.

Author Response

Thank you for your comments. As requested, we have attached Supplementary Figure 4 to the supplementary files with density analyses and the whole membranes for the Western Blots used in the main figures.

Also, during the review process, we noticed some minor errors in Figures 1 and 3 which do not change the conclusions or the results description, so we changed those figures in our revised version.

We hope that you find our manuscript is now suitable for publication.